# Test accuracy of drug and antibody assays for predicting response to antitumour necrosis factor treatment in Crohn's disease: a systematic review and meta-analysis

Karoline Freeman,[1] Sian Taylor-Phillips,[1] Martin Connock,[1] Rachel Court,[1] Alexander Tsertsvadze,[1] Deepson Shyangdan,[1] Peter Auguste,[1] Hema Mistry,[1] Ramesh Arasaradnam,[1,2] Paul Sutcliffe,[1] Aileen Clarke[1]

► Prepublication history and additional material are available. To view these files please visit the journal online (http://dx.doi.org/10.1136/bmjopen-2016-014581).

[1]Warwick Medical School, University of Warwick, Coventry, Warwickshire, UK
[2]Department of Gastroenterology, University Hospital Coventry and Warwickshire, Coventry, UK

**Correspondence to**
Sian Taylor-Phillips; s.taylor-phillips@warwick.ac.uk

## ABSTRACT

**Objective** To present meta-analytic test accuracy estimates of levels of antitumour necrosis factor (anti-TNF) and antibodies to anti-TNF to predict loss of response or lack of regaining response in patients with anti-TNF managed Crohn's disease.

**Methods** MEDLINE, Embase, the Cochrane Library and Science Citation Index were searched from inception to October/November 2014 to identify studies which reported 2×2 table data of the association between levels of anti-TNF or its antibodies and clinical status. Hierarchical/bivariate meta-analysis was undertaken with the user-written 'metandi' package of Harbord and Whiting using Stata V.11 software, for infliximab, adalimumab, anti-infliximab and anti-adalimumab levels as predictors of loss of response. Prevalence of Crohn's disease in included studies was meta-analysed using a random effects model in MetaAnalyst software to calculate positive and negative predictive values. The search was updated in January 2017.

**Results** 31 studies were included in the review. Studies were heterogeneous with respect to the type of test used, criteria for establishing response and loss of response, population examined and results. Meta-analytic summary point estimates for sensitivity and specificity were 65.7% and 80.6% for infliximab trough levels and 56% and 79% for antibodies to infliximab, respectively. Pooled results for adalimumab trough levels and antibodies to adalimumab were similar. Pooled positive and negative predictive values ranged between 70% and 80% implying that between 20% and 30% of both positive and negative test results may be incorrect in predicting loss of response.

**Conclusion** The available evidence suggests that these tests have modest predictive accuracy for clinical status; direct test accuracy comparisons in the same population are needed. More clinical trial evidence from test–treat studies is required before the clinical utility of the tests can be reliably evaluated.

## Strengths and limitations of this study

► This is the first study to summarise predictive accuracy of tests for loss of response to antitumour necrosis factor drugs for managing Crohn's disease, in a clinically relevant manner.
► We included more studies than previous meta-analyses.
► We investigated drug and antibody levels for both infliximab and adalimumab.
► Many of the included studies had a high risk of bias.
► There was insufficient data for subgroup analyses for some types of test.

Sharp & Dohme) and adalimumab (Humira, AbbVie), are well-established second-line or third-line therapies for people with Crohn's disease (CD). Failure to respond during induction therapy and loss of response (LOR) after initial success are widely documented.[1–5] One suggested mechanism for this is the production of antibodies which neutralise the anti-TNFα agents and hasten their clearance from the circulation, thus reducing drug availability. The treatment strategy for LOR is usually to escalate the drug dosage or to shorten the dosage interval. If this fails, a switch to an alternative anti-TNF agent can be tried in order to minimise the influence of anti-drug antibodies directed against the first agent. Another suggested underlying mechanism for LOR is that cytokines other than TNFα may become the major inflammatory agents. This suggestion arises from the observation that some patients have a LOR to anti-TNF despite the presence of therapeutic drug levels and an absence of anti-TNF antibodies. For such patients, the continued use of anti-TNFs may be considered futile and a switch to different

## INTRODUCTION

Antitumour necrosis factor (anti-TNFα) agents, including infliximab (Remicade, Merck

biological therapies or other agents may represent the preferred strategy.

The potential role of anti-TNF antibodies and of subtherapeutic drug levels in LOR has provided the impetus for the development of assays for both anti-TNF drugs and for antibodies, and a plethora of studies using such assays have been produced, exploring the association between either levels of antibodies to anti-TNF agents and clinical response or levels of drugs and clinical response. Studies have measured LOR to the administered anti-TNF agent or failure to regain response after a change in treatment. By dichotomising the outcomes at various detectable levels of drug and of antibodies to anti-TNF, the diagnostic value of these tests in predicting LOR or lack of regaining response has been assessed.

Several authors have meta-analysed studies which have reported the association between levels of antibodies to anti-TNF agents and clinical status.[6–9] These authors have presented pooled relative risk or odds ratio (OR) statistics for clinical state (eg, response or LOR) investigating positive versus negative test result patients (ie, antibodies to anti-TNF agent present or absent), or conversely for test result (positive or negative) in patients with response versus those without response. Although these pooled statistics provide useful information on the association between antibody levels and clinical status, they do not address the question of test accuracy when tests are used as a predictor of patients' clinical response status which is the perspective likely to be adopted by clinicians for patients receiving treatment that may be predicated on test results. Primary studies frequently report test accuracy analysis such as receiver operating characteristic (ROC) curves and test accuracy measures such as sensitivity and specificity. When viewed as diagnostic tests,[10] it becomes possible to perform alternative meta-analysis so as to obtain pooled estimates of test accuracy. The predictive accuracy of such tests is of considerable practical interest. Our objective therefore is to present the meta-analytic results in terms of pooled test accuracy estimates. A particular advantage of this method is that it allows for investigation of the covariance of associations or, from the perspective of a predictive test, the covariance between sensitivity and specificity, thus giving a more complete picture of the value of these tests in clinical practice.

## METHODS
### Search for studies
An iterative procedure was used to develop the initial MEDLINE search, which was subsequently adapted appropriately for other databases and online resources. We searched multiple bibliographic databases including MEDLINE, Embase, the Cochrane Library and Science Citation Index from inception to October/November 2014. Searches of other online resources including trial registries were also undertaken. Full details of the search strategies used, with exact search dates, are provided in the online supplement 1 . Reference lists of included studies and relevant review articles were checked. Citation searches of selected included studies were undertaken. An update of the search was undertaken in January 2017 (see online supplement 2 figure 1 and table 1) .

### Study eligibility criteria
We included studies of patients with CD treated with infliximab or adalimumab. Studies with mixed Crohn's and ulcerative colitis populations were included if the proportion of Crohn's patients was at least 70%. The intervention of interest was a test measuring serum anti-TNFα (infliximab or adalimumab) and/or anti-infliximab or anti-adalimumab antibody levels. Studies reporting clinical status (ie, response or lack of response) as an outcome were eligible for inclusion. The reported results had to allow for cross-tabulation of dichotomous test outcome with clinical status by means of 2×2 tables in order to calculate the diagnostic test accuracy parameters. All primary study designs were included.

### Study selection
Two reviewers independently assessed titles and abstracts for inclusion using a prepiloted form. All potentially relevant publications were retrieved and examined independently. Any disagreements regarding inclusion/exclusion were discussed and resolved with a third reviewer. The study selection process and reasons for exclusion at full text screening level are presented in the PRISMA study flow diagram (see figure 1).

### Quality assessment
Studies were quality assessed using a modified QUADAS-2 checklist.[11] Items included were method of patient selection, blinding of index test results, exclusion of uninterpretable test results from 2×2 table data and method of assessment of clinical status (the reference case).

### Evidence synthesis and statistical methods
Patient numbers within extracted 2×2 data tables were used to generate Forest plots of paired sensitivity and specificity (accompanied by 95% CIs) using Review Manager (RevMan V.5.1; Nordic Cochrane Centre, Copenhagen, Denmark) for four different tests: (1) infliximab levels as predictor of loss of or lack of regaining response, (2) antibodies to infliximab as predictor of loss of or lack of regaining response, (3) adalimumab levels as predictor of loss of or lack of regaining response and (4) antibodies to adalimumab as predictor of loss of or lack of regaining response. Hierarchical/bivariate[12] meta-analysis was undertaken with the user-written 'metandi' package of Harbord and Whiting[13] using Stata V.11 software. Positive and negative predictive values were calculated[14] at the pooled prevalence of LOR in the test population. Prevalence was meta-analysed using a random effects model in MetaAnalyst software.[15] For meta-analyses which incorporated 10 or more studies, we examined the risk of publication bias (see online supplement 3) mindful of the caveats relating to this in diagnostic test accuracy studies.[16]

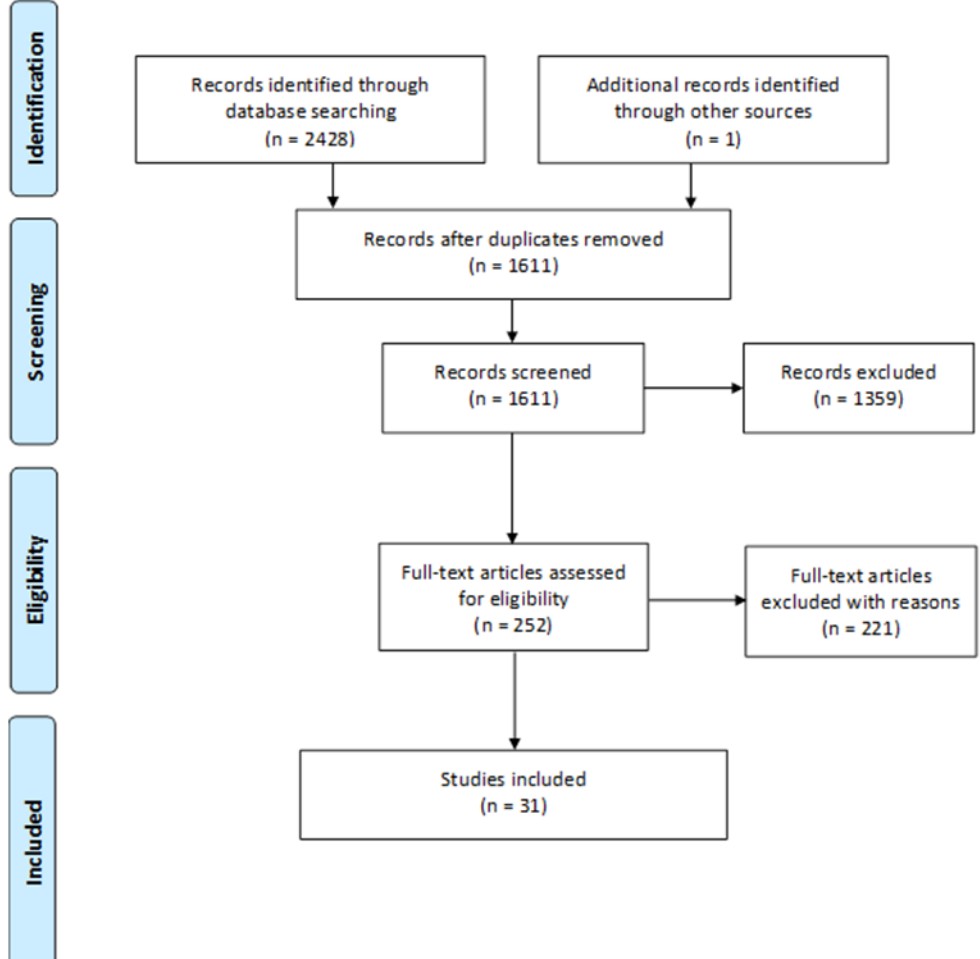

**Figure 1** PRISMA study flow diagram.

The protocol for this review was registered on PROS-PERO 2014:CRD42014015278. The full protocol is included in the online supplementary appendix 1.

## RESULTS

We identified 2429 records of which 31 were eligible for inclusion (see online supplement 4 table 1 for excluded studies with reason). Of these, 24 were full-text reports and 7 were conference abstracts. The PRISMA flow diagram is detailed in figure 1. Eleven of the 31 studies examined infliximab trough levels, 20 examined levels of antibodies to infliximab and five and six studies, respectively, investigated adalimumab levels and antibodies to adalimumab (table 1). The range of anti-TNF cut-offs used for the dichotomisation of test outcomes is illustrated in the online supplement 5 tables 1–3) . The risk of bias of studies varied. The greatest threat to validity was high risk of bias in patient selection, for example, studies did not enrol a consecutive or randomly selected patient group. This was present in nearly 80% of included studies (see online supplement 6 table 1 and figure 1).

The studies were heterogeneous with respect to the type of test used (eg, commercial or in-house ELISA, radioimmunoassay (RIA), homogeneous mobility shift assay (HMSA)), criteria for establishing response or lack of regaining response (eg, use of the Crohn's Disease Activity Index score or the physician's global assessment score) and population examined (responders or patients with secondary loss of response). Sensitivity and specificity pairs are summarised in figure 2 for antibodies to anti-TNF and figure 3 for anti-TNF trough levels.

The paired forest plots show that sensitivity and specificity of using anti-TNFs or antibodies produced against anti-TNFs to predict response or LOR vary greatly among studies with sensitivity revealing generally greater variation. Sensitivity analysis suggests that assay type may explain some of the variation in results between studies of anti-infliximab antibodies; however, there was considerable heterogeneity between numerous study covariates (population, assay type, response criterion), and we do not know whether these might fully explain the large differences in results between studies.

### Infliximab trough level tests for LOR or lack of regaining response

Of 11 included studies, 2 were reported only as abstracts (Ben-Basset et al[17] and Yanai et al[18]). The meta-analysis (figure 4) yielded a pooled summary point of 66%

**Table 1**   Major features of studies included for hierarchical meta-analyses

| Study | Drug | Diagnosis | Response/LOR | Test | Response measure |
|---|---|---|---|---|---|
| Trough antibodies to infliximab as predictor of loss of or lack of regaining response | | | | | |
| Ben-Horin et al[62] | IFX ADA | IBD~0.9 CD | LOR | ELISA | PJ |
| Candon et al[63] | IFX | CD | LOR | ELISA | UC |
| Pariente et al[64] | IFX | CD and UC | LOR | ELISA | PJ or HBI |
| Baert et al[65] | IFX | IBD~0.8 CD | LOR | HMSA | PJ |
| Vande Casteele et al[24] | IFX | IBD~0.70 CD | LOR | HMSA | CRP TC |
| Ainsworth et al[66] | IFX | CD | LOR | RIA | PJ |
| Steenholdt et al[26] | IFX | CD | LOR | RIA | CDAI |
| Farrell et al[67] | IFX | CD | Resp | ELISA | PJ |
| Hanauer et al[25] | IFX | CD | Resp | ELISA | CDAI |
| Imaeda et al[27] | IFX | CD | Resp | ELISA | CDAI |
| Kong et al[19] abstract | IFX | IBD~0.83 CD | Resp | ELISA | PJ |
| Kopylov et al[40] | IFX | CD | Resp | ELISA | PJ |
| Marzo et al[20] abstract | IFX | NR | Resp | ELISA | CDAI |
| Nagore et al[21] abstract | IFX | IBD~0.86 CD | Resp | ELISA | PJ |
| Steenholdt et al[68] | IFX | CD | Resp | ELISA | PJ |
| Bodini et al[22] abstract | IFX | CD | Resp | HMSA | HBI |
| Vande Casteele et al[24] | IFX | IBD~0.70 CD | Resp | HMSA | CRP TC |
| Steenholdt et al[69] | IFX | CD | Resp | RIA | PJ ST |
| Ben-Horin et al[70] | IFX | IBD~0.82 CD | Resp | NR | ST |
| Dauer et al[23] abstract | IFX | IBD~0.83 CD | Resp | NR | PJ |
| Trough antibodies to adalimumab as predictor of loss of or lack of regaining response | | | | | |
| Imaeda et al[28] | ADA | CD | Resp | ELISA | CRP |
| Mazor et al[71] | ADA | CD | Resp | ELISA | PJ+CRP |
| Roblin et al[72] | ADA | CD | Resp | ELISA | CDAI |
| Frederiksen et al[73] | ADA | IBD | Resp | RIA | PJ BM |
| West et al[74] | ADA | CD | Resp | RIA | PJ |
| Ben-Horin et al[62] | IFX ADA | IBD~0.9 CD | LOR | ELISA | SA |
| Infliximab trough level as predictor of loss of or lack of regaining response | | | | | |
| Ainsworth et al[66] | IFX | CD | LOR | RIA | PJ |
| Steenholdt et al[26] | IFX | CD | LOR | RIA | CDAI |
| Bortlik et al[75] | IFX | CD | Resp | ELISA | PJ |
| Cornillie et al[76] | IFX | CD | Resp | ELISA | CDAI |
| Hibi et al[77] | IFX | CD | Resp | ELISA | CDAI |
| Imaeda et al[27] | IFX | CD | Resp | ELISA | CDAI |
| Kopylov et al[40] | IFX | CD | Resp | ELISA | PJ |
| Yanai et al[18] abstract | IFX | CD | Resp | ELISA | PJ |
| Ben-Basset et al[17] abstract | IFX | IBD~0.93 CD | Resp | HMSA | HBI |
| Steenholdt et al[69] | IFX | CD | Resp | RIA | PJ |
| Maser et al[78] | IFX | CD | Resp | ELISA | HBI |
| Adalimumab trough level as predictor of loss of or lack of regaining response | | | | | |
| Chiu et al[39] | ADA | CD | LOR | ELISA | CDAI |
| Imaeda et al[28] | ADA | CD | Resp | ELISA | CRP |
| Mazor et al[71] | ADA | CD | Resp | ELISA | PJ+CRP |
| Roblin et al[72] | ADA | CD | Resp | ELISA | CDAI |
| Frederiksen et al[73] | ADA | IBD | Resp | RIA | PJ BM |

ADA, adalimumab; CD, Crohn's disease; CDAI, Crohn's Disease Activity Index score; CRP, C reactive protein level; Diagnosis, study patient population; HBI, Harvey Bradshaw Index score; HMSA, homogeneous mobility shift assay; IBD, inflammatory bowel disease; IFX, infliximab; LOR, patients with loss of response; NR, not reported; PJ, physicians' judgement; PJ BM, physicians' judgement and biological measure; Resp, responding patients; Response measure, method used for defining clinical response; RIA, radioimmunoassay; SA, switch anti-TNF; ST, stop anti-TNF; TC, treatment change.

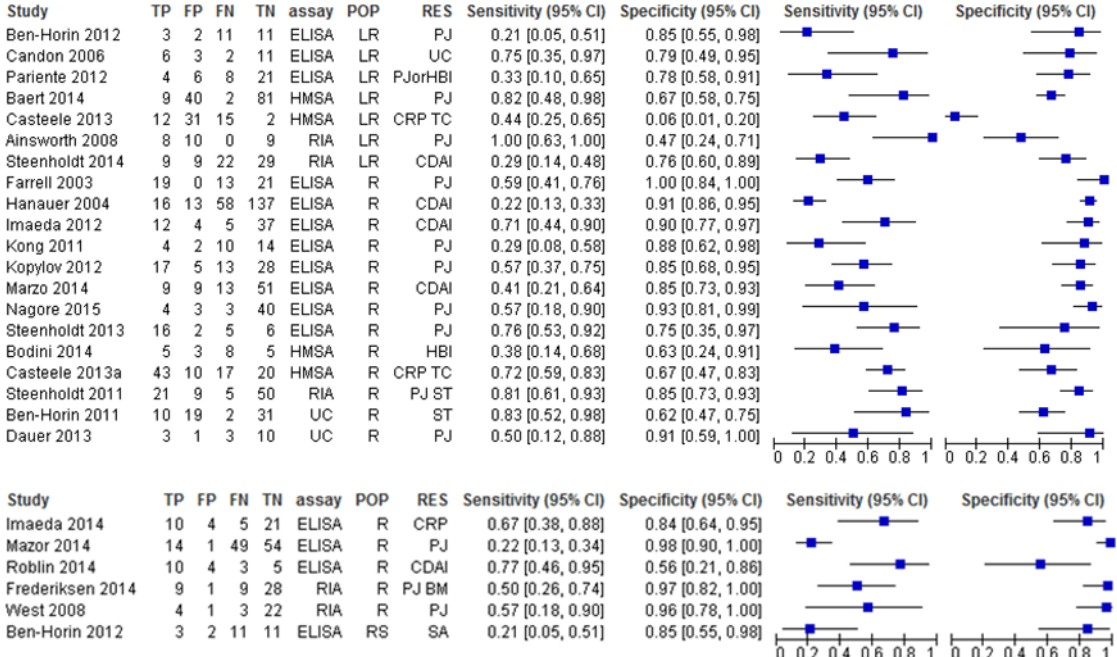

**Figure 2** Paired forest plots for anti-TNF antibody levels for predicting loss of response or failure to regain response to infliximab (top) and adalimumab (bottom). CDAI, Crohn's Disease Activity Index score; CRP, C reactive protein level; HBI, Harvey Bradshaw Index score; HMSA, homogeneous mobility shift assay; LR, patients with loss of response; PJ, physicians' judgement; PJ BM, physicians' judgement and biological measure; POP, study patient population; R, patients with response; RES, criterion for determining clinical response; RIA, radioimmunoassay; RS, restart anti-TNF after drug holiday; SA, switch anti-TNF; ST, stop anti-TNF therapy; TC, treatment change; UC, unclear.

sensitivity and 81% specificity (other test accuracy statistics are summarised in the online supplement 3). Sensitivity analysis in which only studies of responder populations were included generated very similar results as did analysis that only included studies with ELISA tests.

### Antibodies to infliximab tests for LOR or lack of regaining response

Of 20 included studies, 5 were reported as abstracts.[19–23] Sensitivity and specificity pairs are summarised in

figure 5. The pooled summary points for sensitivity and specificity were 56% and 79%, respectively (figure 5). Only minor differences were introduced in the test accuracy outcomes (eg, 60% and 81% for sensitivity and specificity, respectively) in a sensitivity analysis when two influential studies were omitted from the analysis.[24 25] Sensitivity analyses in which only responder studies were included had little effect. Sensitivity analysis in which only ELISA studies were included showed an improvement in specificity at the expense of sensitivity and a

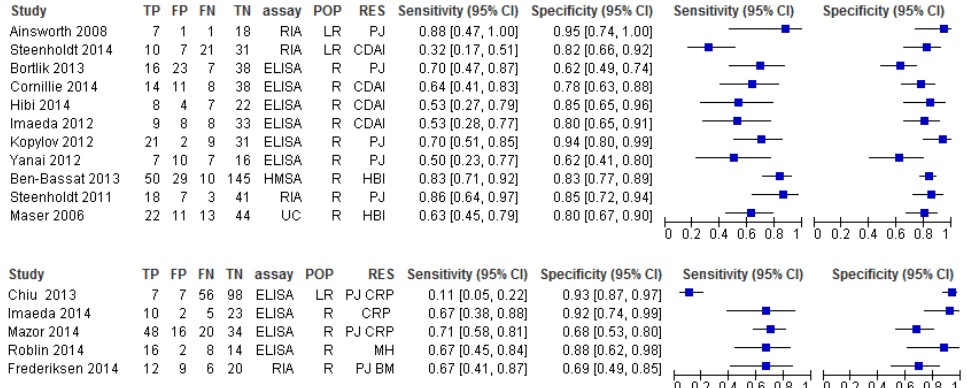

**Figure 3** Paired forest plots for trough anti-tumour necrosis factor levels for predicting loss of response or failure to regain response to infliximab (top) and adalimumab (bottom). CDAI, Crohn's Disease Activity Index score; CRP, C reactive protein level; HBI, Harvey Bradshaw Index score; HMSA, homogeneous mobility shift assay; LR, patients with loss of response; MH, mucosal healing; PJ, physicians' judgement; PJ BM, physicians' judgement and biological measure; POP, study patient population; R, patients with response; RES, criterion for determining clinical response; RIA, radioimmunoassay; UC, unclear.

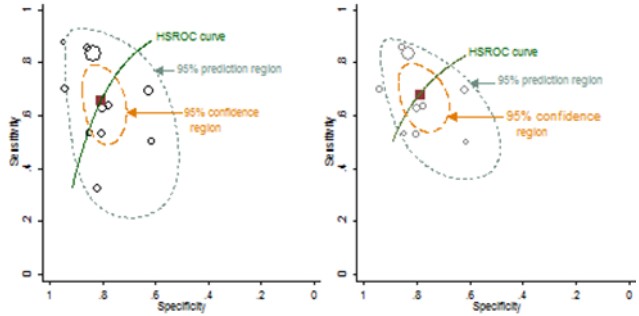

**Figure 4** Hierarchical meta-analysis of trough infliximab levels for predicting loss of response or failure to regain response. Left, all 11 studies; right, responder studies only (n=9). The square symbol represents the summary point estimate on the hierarchical summary receiver operating characteristic (HSROC) curve.

reduction in the heterogeneity of specificity measurements (figure 5).

### Adalimumab or anti-adalimumab antibody levels as tests for LOR or lack of regaining response

Far fewer studies of adalimumab-treated patients were available compared with infliximab (table 1). Meta-analysis of patients treated with adalimumab yielded slightly lower test accuracy statistics with wider uncertainty

around them compared with those found for infliximab studies (see online supplement 7 table 1 and figure 1).

### Combined assessment of anti-TNF levels and antibodies to anti-TNF

Three independent studies reported both drug and antibody test results by individual in relation to the individual's clinical status, response/LOR[26 27] or regaining response/not regaining response.[28] These studies allowed calculation of the number of patients in each of the two clinical states distributed to each of the four possible combinations of test result.[26–28] The results summarised in tables 2 and 3 indicate the probability of LOR to anti-TNF, and table 4 summarises the probability of not regaining response to infliximab according to each possible test result category. These test results are reasonably similar to those from our meta-analysis of single test studies. This comparison should be viewed in the light of the considerable uncertainty which exists because of the small number of studies measuring both drug and antibody levels in the same individuals and their small size.

### Predictive values of drug and antidrug antibody tests for LOR or failure to regain response

In the Cochrane Handbook for Systematic Reviews of Diagnostic Test Accuracy, Bossuyt et al (2013)[14] suggest that predictive values are more widely and readily

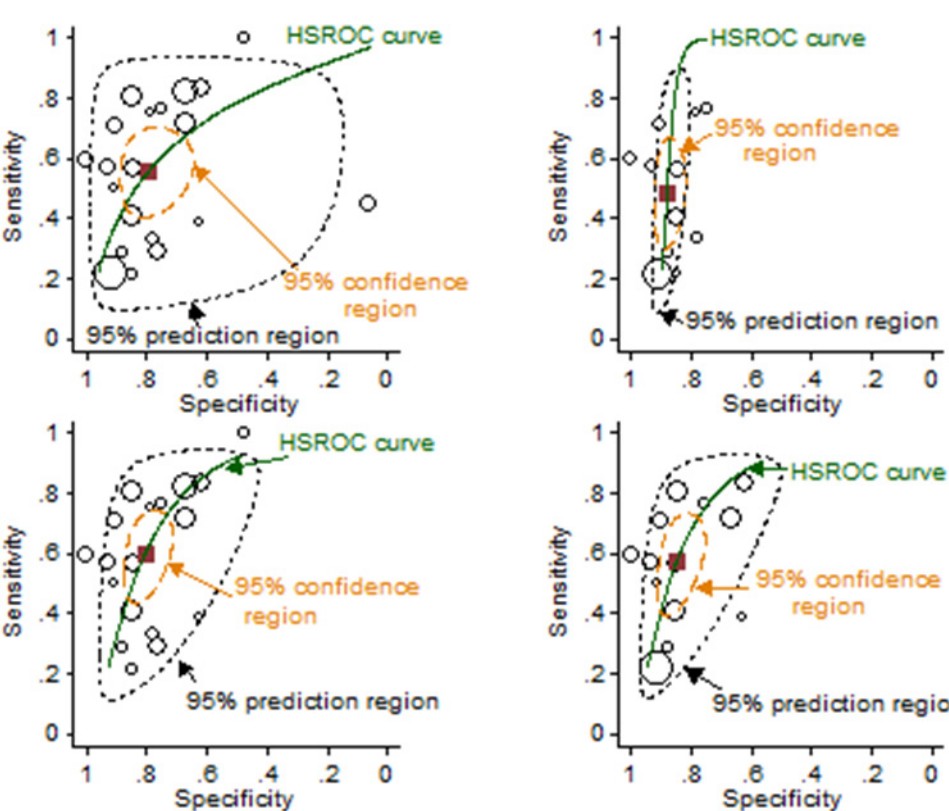

**Figure 5** Hierarchical meta-analysis of trough levels of antibodies to infliximab for predicting loss of response or failure to regain response. Top left, all 20 studies; top right, ELISA studies only (n=9); lower left, all studies minus two influential studies (n=18)[26 66]; lower right, responder studies only (n=13). The square symbol represents the summary point estimate on the hierarchical summary receiver operating characteristic (HSROC) curve.

**Table 2** Combined assessment of adalimumab and anti-adalimumab levels for responders receiving adalimumab

| Imaeda et al[28] | ADAbs+ | ADAbs− | Total | Population and anti-TNFα therapy; tests |
|---|---|---|---|---|
| Anti-TNFα− | LOR=8 | LOR=2 | LOR=10 | Responders on adalimumab maintenance; ELISA. Prevalence of LOR=37.5% |
|  | RESP=0 | RESP=2 | RESP=2 |  |
| Anti-TNFα+ | LOR=2 | LOR=3 | LOR=5 |  |
|  | RESP=4 | RESP=19 | RESP=23 |  |
| Total | LOR=10 | LOR=5 | LOR=15 |  |
|  | RESP=4 | RESP=21 | RESP=25 |  |

The probability of a patient returning each of the four possible test result combinations was:
ADAbs+/anti-TNFα−, 0.200; ADAbs+/anti-TNFα+, 0.150; ADAbs−/anti-TNFα−, 0.10; ADAbs−/anti-TNFα+, 0.550.
The probabilities of losing response according to the category of test result were 1.00, 0.333, 0.500 and 0.136, respectively.
ADAbs, antidrug antibodies; ; LOR, loss of response; RESP, responders; ; TNF, tumour necrosis factor.

appreciated than alternative test accuracy statistics such as sensitivity and specificity. Negative and positive predictive values vary according to prevalence of the condition being tested for (in this case lack of response). We have meta-analysed the prevalence across the included studies and used this with its 95% CI as a guide to the approximate prevalence in which the tests would be performed in practice. The predictive values for each type of test across the relevant prevalence ranges are summarised in figure 6. As prevalence increases, positive predictive value increases and negative predictive value decreases.

Although pooled prevalence varies somewhat among the four collections of studies, the resulting positive and negative predictive values are similar and range between about 70% and 80% implying that between 20% and 30%

**Table 3** Combined assessment of infliximab and anti-infliximab for responders receiving infliximab

| Imaeda et al[27] | ADAbs+ | ADAbs− | Total | Population and anti-TNFα therapy; tests |
|---|---|---|---|---|
| Anti-TNFα− | LOR=9 | LOR=0 | LOR=9 | Responders on infliximab maintenance; ELISA. Prevalence of LOR=29.3% |
|  | RESP=1 | RESP=7 | RESP=8 |  |
| Anti-TNFα+ | LOR=3 | LOR=5 | LOR=8 |  |
|  | RESP=3 | RESP=30 | RESP=33 |  |
| Total | LOR=12 | LOR=5 | LOR=17 |  |
|  | RESP=4 | RESP=37 | RESP=41 |  |

The probability of a patient returning each of the four possible test result combinations was:
ADAbs+/anti-TNFα−, 0.172; ADAbs+/anti-TNFα+, 0.103; ADAbs−/anti-TNFα−, 0.121; ADAbs−/anti-TNFα+, 0.603.
The probabilities of losing response according to the category of test result were 0.900, 0.500, 0.000 and 0.143, respectively.
ADAbs, antidrug antibodies; ; LOR, loss of response; RESP, responders; TNF, tumour necrosis factor.

of positive and negative test results are likely to be incorrect.

In January 2017, we updated our included studies by searching all citations of, and included studies in, five relevant systematic reviews (see online supplement 2 figure 1).[6 7 29–31] After removal of duplicates and the application of our inclusion criteria, this yielded three[32–34] and five[33 35–38] additional studies, respectively, for trough infliximab and trough adalimumab levels (see online supplement 8 table 1). Addition of the former to our meta-analysis had almost no influence on our estimates of test accuracy (see online supplement 8 figures 1 and 2 and table 2); the addition of the adalimumab studies to our meta-analysis also had very little influence on our estimates of test accuracy except a modest reduction in their uncertainty despite doubling the number of available studies (see online supplement 8 figures 1 and 3 and table 3).

## DISCUSSION
The meta-analysis results indicate that the accuracy of tests for predicting lack of response was moderate and that about 20%–30% of both positive and negative test results are likely to be incorrect, with large unexplained heterogeneity between studies. The number of studies on patients treated with adalimumab was too small to draw firm conclusions, but the available evidence suggests similar performance to the tests for infliximab and for antibodies to infliximab.

The sensitivity analyses indicated that much of the variation seen in the forest plots and ROC space could not be explained by our measures of test type and population. Test performance is dependent on cut-offs used for anti-TNF and antibodies to anti-TNF agents and on the time of testing. However, this was not investigated in sensitivity analyses as cut-offs vary by test type as well as within different types of tests, and an agreed cut-off that is transferable between studies and populations has yet to be identified. Furthermore, time of testing was not investigated as all but one study[39] reported that anti-TNFs levels considered in the studies were trough levels.

Updating the searches found an extra seven studies; however, these made no meaningful difference to the test accuracy estimates. The study designs were largely similar to those in the previous studies. However, there appears to have been a recent waning of interest in antidrug antibodies, possibly attributable to publication of studies indicating their transitory and varying persistence during treatment, while interest in endoscopic healing as an outcome appears to have increased. Additional single arm test accuracy studies may not add significant further understanding in this field. Of more value would be head-to-head test accuracy comparisons in the same population and studies integrating drug levels with other predictive factors to enable more accurate predictions of LOR.

**Table 4** Combined assessment of infliximab and anti-infliximab for people with loss of response receiving infliximab

| Steenholdt et al[26] | ADAbs+ | ADAbs− | Total | Population and anti-TNFα therapy; tests |
|---|---|---|---|---|
| Anti-TNFα− | NOR=8 | NOR=2 | NOR=10 | Failure on infliximab, continued failure or gain of response at 12 weeks; RIA. Prevalence of NOR=44.9% |
| | RESP=6 | RESP=1 | RESP=7 | |
| Anti-TNFα+ | NOR=1 | NOR=20 | NOR=21 | |
| | RESP=3 | RESP=28 | RESP=31 | |
| Total | NOR=9 | NOR=22 | NOR=31 | |
| | RESP=9 | RESP=29 | RESP=38 | |

The probability of a patient returning each of the four possible test result combinations was:
ADAbs+/anti-TNFα−, 0.203; ADAbs+/anti-TNFα+, 0.058; ADAbs−/anti-TNFα−, 0.00043; ADAbs−/anti-TNFα+, 0.696.
The probabilities of failing to gain a response according to category of test result were 0.571, 0.250, 0.667 and 0.417, respectively.
ADAbs, antidrug antibodies; LOR, loss of response; RESP, responders; NOR, no regain of response; TNF, tumour necrosis factor.

Our meta-analyses included studies using different tests for measuring levels of anti-TNF agents and antibodies to anti-TNFs. Although RIA and HMSA tests were used in some of our included studies, the bulk of the tests employed were ELISA tests (26/42, 62%) encompassing various commercial ELISA kits and ELISAs developed 'in house' by investigators. Several full publications and abstracts have addressed the issue of whether different test methods (eg, solid phase ELISAs, liquid phase assays such as RIA or HMSA) deliver the same quantitative estimates of drug and antibody levels in patient samples.[22 24 26–28 40–59] Because there is no consensus about what constitutes a gold standard test, it is difficult to draw conclusions from these studies other than that some differences in performance have been documented. Interestingly, the observed

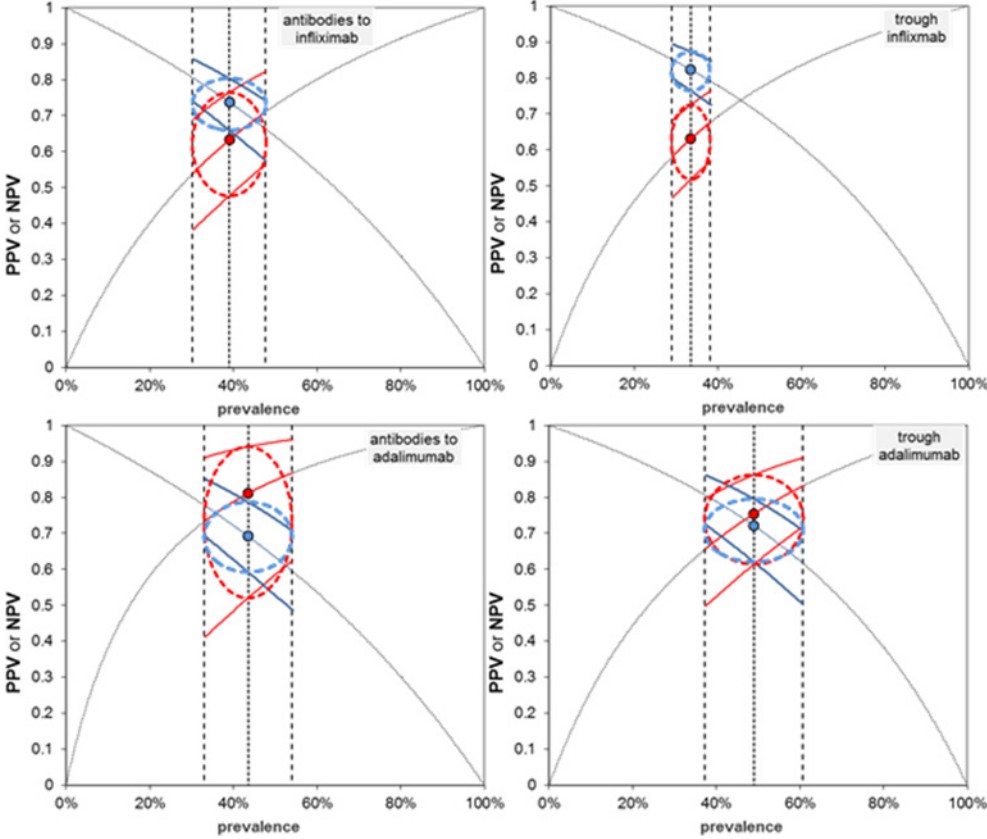

**Figure 6** Positive predictive value (PPV) and negative predictive value (NPV) according to the prevalence of lack of response using the pooled summary receiver operating characteristic (sROC) model estimates of sensitivity and specificity. Data points, PPV and NPV at sROC pooled sensitivity and specificity and pooled prevalence; vertical dashed lines, pooled prevalence and 95% CIs; thick curves, PPV and NPV at upper and lower CIs for sensitivity and specificity across the pooled prevalence and its 95% CI. The dashed line ellipses encompass predictive values determined from 95% CIs of prevalence and 95% CI for PPV and NPV at the point prevalence estimate.

variation in our meta-analysis could not be explained by the different tests used.

Although the accuracy of the tests for predicting lack of response was found to be moderate, this does not necessarily mean they must lack clinical utility. However, clinicians are likely to be interested in a combined assessment of anti-TNF levels and antibodies to anti-TNF, for which limited accuracy data are available.[26–28] Diagnostic tests may alter clinical decisions and actions, so evidence beyond test accuracy is required to evaluate clinical value.[60] Such evidence is best obtained in randomised trials (ie, test and treat investigations), but this is currently sparse.[60]

Two recent randomised controlled trials (RCTs) have compared clinical outcomes between patients whose treatment was directed by algorithms informed by tests for infliximab and/or antibodies to infliximab versus patients who received treatment uninformed by testing.[26 61] In the TAXIT trial,[61] patients with inflammatory bowel disease responding to infliximab had their dose regimen optimised according to a test algorithm with the aim to bring patients within the therapeutic range and prevent LOR. However, after randomisation to clinically based or test-based dosing, no clinical benefit was observed for patients with CD at 1 year. Steenholdt et al (2014)[26] investigated patients who had lost response to infliximab, using a test algorithm to predict the reason for LOR and adjust treatment accordingly. In this equivalence study, no difference in clinical benefit was observed for the test-algorithm group relative to the control group who were prescribed dose intensification. It is notable in this study that for many patients (14/33; 42%), clinicians failed to implement the test-algorithm directive, implying that they may have lacked confidence in the test results or that they considered other factors of overriding importance, as pointed out by Ferrante di Ruffano et al (2012).[60] Such phenomena (lack of equipoise) complicate assessments of test value. Both of these RCTs reported cost savings in the test-algorithm arm associated with reduced use of infliximab.

This is the first meta-analysis of predictive accuracy of these tests and offers an alternative perspective to earlier meta-analyses. We were able to include more studies than in earlier meta-analyses and have looked at both drug tests and tests for antidrug antibodies and have included studies of patients receiving either infliximab or adalimumab therapies. There was significant heterogeneity between studies, including in the test, outcome measurement and findings, making clinical interpretation difficult.

The meta-analysis results should be viewed with some caution because of the high risk of bias in many of the included studies and because the lack of sufficient numbers of studies precluded subgroup meta-analyses of some types of test (eg, RIA, HMSA).

## CONCLUSIONS

The available evidence suggests that these tests have modest predictive accuracy for clinical status and that about 20%–30% of test results would be likely to be incorrect. However, higher quality head-to-head test accuracy studies are required to enable differentiation between different types of tests and cut-offs, with consistent outcome measurement in the same population. In published trials, the tests have been used for adjusting dose or treatment of patients whose clinical status has already been defined by other criteria. More clinical trial evidence from test–treat studies is required before the clinical utility of the tests can be reliably evaluated.

**Contributors** KF coordinated the review. RC developed the search strategy and undertook searches. KF, ST-P, MC, AT, DS and PS conducted the clinical effectiveness systematic review, this included: screening and retrieving papers, assessing against the inclusion criteria, appraising the quality of papers and abstracting data from papers for synthesis. MC conducted the data analysis. AC obtained funding, provided project management and methodological advice. RA provided clinical comment and guidance. All authors were involved in writing draft versions of the paper and approved the final version submitted.

**Funding** This work was commissioned by the NIHR HTA Programme as project number 14/69/03. AC and STP are partly supported by the National Institute for Health Research (NIHR) Collaboration for Leadership in Applied Health Research and Care West Midlands at the University Hospitals Birmingham NHS Foundation Trust.

**Competing interests** None declared.

**Provenance and peer review** Not commissioned; externally peer reviewed.

**Data sharing statement** All data are available from authors on request.

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
