## [Reviewer comments · BMJ Open]

ARTICLE DETAILS

TITLE (PROVISIONAL)	Test accuracy of drug and antibody assays for predicting response to anti-Tumour Necrosis Factor treatment in Crohn's disease: a systematic review and meta-analysis
AUTHORS	Freeman, Karoline; Taylor-Phillips, Sian; Connock, Martin; Court, Rachel; Tsertsvadze, Alexander; Shyangdan, Deepson; Auguste, Peter; Mistry, Hema; Arasaradnam, Ramesh; Sutcliffe, Paul; Clarke, Aileen

VERSION 1 - REVIEW

REVIEWER	Roger Harbord Met Office, UK
REVIEW RETURNED	20-Nov-2016

GENERAL COMMENTS	In general I found the statistical methods to be appropriate and adequately described, the results well-reported and the discussion and conclusion justified by the results, but I have some specific points: Abstract: "meta-analytic results for sensitivity and specificity" - I would prefer to see something like "meta-analytic summary point estimates for sensitivity and specificity" with mention of the substantial heterogeneity. Page 2 lines 45-6: "loss of response to Crohns disease" - surely loss of response to anti-TNF drugs for managing Crohns disease? Methods: No mention is made of how a single pair of sensitivity and specificity was selected for studies reporting values at more than one cut-off (as shown in Supplement 2). Page 5 line 13: Appendix 5 should read Supplement 5. Results: Page 5 line 33 "the greatest threat to validity was high risk of bias in patient selection" - it would be useful to add a few words to summarise the reasons for this. Page 5 lines 48-9: " None of the presented covariates (population, assay type, response criterion) appear to explain the observed variation." I cannot see where these covariates are presented. I am unclear how the it was judged whether each covariate explained the variation - did this used statistical methods or was it done by eye? In addition page 7 lines 21-3 appear to indicate that there *was*
---

heterogeneity by assay type, namely that in ELISA studies there was an improvement in specificity at the expense of sensitivity.

Table 1:

The initialisms NR and HMSA should be included in the key at the bottom. It would seem preferable if the ordering of tests and studies was consistent between Table 1 and Figs 2-3. (The current ordering in Table 1 appears alphabetical apart from the last section, "Trough antibodies to Adalimumab...".)

Page 7: It might be preferable to report summary sensitivities and specificities as percentages, as in the abstract.

Page 7 lines 21-3: In addition to the improvement in specificity at the expense of sensitivity in the ELISA studies alone, it seems worth mentioning the large reduction in the heterogeneity in specificity evident in Figure 5.

Page 7 lines 45-6: It does not seem sensible to use the 95% CI from the meta-analysis as an estimate of the range of likely value across which the test would be performed in practice, as the 95% CI merely indicates the uncertainty in the summary estimate from the meta-analysis. A prediction interval (which will be wider unless the estimated between-study variance tau-squared is zero) would seem more appropriate, or simply the range of prevalences from the included studies.

Discussion:

Page 8 lines 14-15: "The sensitivity analyses indicated that the variation seen in the Forest plots and ROC space could not be explained by test type" - this appears inconsistent with page 7 lines 21-3 and Figure 5 in which restricting to ELISA studies does explain some of the heterogeneity.

Page 8 lines 42-5: " clinicians are likely to be interested in a combined assessment of anti-TNF levels and antibodies to anti-TNF, for which limited accuracy data is available.[21 25 43]". If this is likely to be of interest to clinicians, would it be worth giving a brief narrative summary of the results of these three studies?

Figure 1: The PRISMA diagram reports the number of "Full-text articles excluded with reasons", but I cannot see where the reasons were reported - if in one of the Supplements, this should be noted in the caption.

Figures 2-3: Legends should start (or include) "Paired forest plots for..." and indicate which group of studies is Infliximab and which Adalimumab.

Figures 4-5: The distinction between the separate panels should be explained in the legends.

Figure 6: The caption needs expanding to explain the solid and dashed red and blue lines.

Supplement 4 Table S4: Please indicate somewhere (e.g. in a footnote to the table) which studies were considered outliers and

	omitted from the second meta-analysis of trough level of antibodies to Infliximab. Figure S1: In the legend, "Chui" should be "Chiu". A reason should be given for omitting the study of Mazor (particularly large/influential?)
--	--

REVIEWER	Remo Panaccione University of Calgary, Canada
REVIEW RETURNED	01-Dec-2016

GENERAL COMMENTS	Although this is an excellent attempt to summarize the validity, sensitivity, and specificity of TDM primarily in patients who lose response to anti-TNF (namely infliximab and adalimumab) the results are not clinically relevant primarily because of the significant heterogeneity in the assays used(cut-off, sensitivity for antibody detection, range of levels) as well as the heterogeneity in the corresponding clinical definitions. Even this type of meta-analysis can't correct for this and therefore this work adds little to what is already known in the field.
---

REVIEWER	Luisa Guidi MD PhD Dept. Gastroenterology Fondazione Policlinico Universitario A. Gemelli, Rome, Italy
REVIEW RETURNED	26-Dec-2016

GENERAL COMMENTS	The systematic review and meta-analysis is correctly designed and reported. It addresses a very significant topic in the field of inflammatory bowel disease care, that is the role of therapeutic drug monitoring of anti-TNF drugs. I have much appreciated that the Authors have accounted for all the more relevant issues in the field, the most important being the heterogeneity of assay methods and the wide range of different cut-off levels employed in the different studies. The main result of the study, that the positive and negative predictive values of these assays for loss of response or failure to regain response range between 70 and 80%, is consistent with the current opinion of the experts in the field. However, this is not the only way to look at these data. The combined assay of anti-TNF and anti drug antibodies adds value, as well as the analysis of their role as a guide to manage the loss of response in a more rational way as compared to the empirical one. This could also allow a significant cost reduction. These arguments are correctly included in the discussion of this paper, as well as the acknowledgement of the need of test-treat studies to define the real clinical utility of these tests.
--

REVIEWER	Dr Gordon W. Moran Clinical Associate Professor and Honorary Consultant Gastroenterologist Academic Program Director for Gastroenterology Inflammatory Bowel Disease Clinical Lead NIHR Biomedical Research Unit in Gastrointestinal and Liver Diseases at Nottingham University Hospitals NHS Trust and The University of Nottingham
-----------------	--

	D1406, West Block, Queen's Medical Centre Nottingham University Hospitals NHS Trust
REVIEW RETURNED	26-Dec-2016

GENERAL COMMENTS	Well written paper and overall will be a valuable contribution within this field. There are some issues to target here first: A few comments that may be passed on to the authors.  1. The search was done in 2014. Can the authors justify this delay. A vast array of work has been published since. 2. Most commercial kits will not provide an antibody level if a trough level is available. I see the authors have done a sub-analysis excluding ELISAs from the analysis to investigate the effect of this. this limitation should be discussed. 3. Why no data on ulcerative colitis has been included in this study. I appreciate the data is poor here and no trough levels has been found to be predictive. 4. Are the TNF levels studied here all trough levels or are they at different stages in the dosing cycle - this needs to be discussed 5. Methodology is commendable and nil to discuss here. 6. The authors have described response as symptoms based - CDAI or PGA. I appreciate this is just reflective of the paucity of data published. This is a major limitation. The most predictive outcomes in IBD is mucosal healing and the dichotomous should be based on that. Suggest either doing a sub-analysis on such studies or discuss this major limitation in the literature.
---

VERSION 1 – AUTHOR RESPONSE

Reviewer	Author response
Reviewers' Comments to Author: Reviewer: 1 Reviewer Name: Roger Harbord Institution and Country: Met Office, UK Competing Interests: None declared	
In general I found the statistical methods to be appropriate and adequately described, the results well-reported and the discussion and conclusion justified by the results, but I have some specific points:	Thank you
Abstract:"meta-analytic results for sensitivity and specificity" - I would prefer to see something like "meta-analytic summary point estimates for sensitivity and specificity" with mention of the substantial heterogeneity.	Changed to "Studies were heterogeneous with respect to type of test used, criteria for establishing response and loss of response, population examined, and results. Meta-analytic summary point estimates for sensitivity and specificity ..." as advised
Page 2 lines 45-6: "loss of response to Crohns disease" - surely loss of response to anti-TNF drugs for managing Crohns disease?	Changed to "This is the first study to summarise predictive accuracy of tests for loss of response to anti-TNF drugs for managing Crohns disease, in a clinically relevant manner"
Methods:	

No mention is made of how a single pair of sensitivity and specificity was selected for studies reporting values at more than one cut-off (as shown in Supplement 2).	In the majority of papers only a single cut-point was reported, where 2x2 data was provided for more than one cut-point the authors selected the point of maximum performance when valuing sensitivity and specificity equally.
Page 5 line 13: Appendix 5 should read Supplement 5.	Corrected
Results:	
Page 5 line 33 "the greatest threat to validity was high risk of bias in patient selection" - it would be useful to add a few words to summarise the reasons for this.	Changed to "The greatest threat to validity was high risk of bias in patient selection, for example studies did not enrol a consecutive or randomly selected patient group. This was present in nearly 80% of included studies (Supplement 3)."
Page 5 lines 48-9: "None of the presented covariates (population, assay type, response criterion) appear to explain the observed variation." I cannot see where these covariates are presented. I am unclear how it was judged whether each covariate explained the variation - did this use statistical methods or was it done by eye? In addition page 7 lines 21-3 appear to indicate that there "was" heterogeneity by assay type, namely that in ELISA studies there was an improvement in specificity at the expense of sensitivity.	This was not done by statistical analysis and we agree the sentence needs changing. The sentence has been reworded as follows: : "Sensitivity analysis suggests assay type may explain some of the variation in results between studies of anti-infliximab antibodies, however there was considerable heterogeneity between numerous study covariates (population, assay type, response criterion) and we do not know whether these might fully explain the large differences in results between studies."
Table 1:	
The initialisms NR and HMSA should be included in the key at the bottom. It would seem preferable if the ordering of tests and studies was consistent between Table 1 and Figs 2-3. (The current ordering in Table 1 appears alphabetical apart from the last section, "Trough antibodies to Adalimumab...".)	Added "HMSA= Homogenous Mobility Shift Assay; NR=Not Reported; Figures 2 and 3 are ordered by population then test, we have re-ordered table 1 to match. We have also rearranged the text to match the same order.
Page 7: It might be preferable to report summary sensitivities and specificities as percentages, as in the abstract.	Amended as suggested
Page 7 lines 21-3: In addition to the improvement in specificity at the expense of sensitivity in the ELISA studies alone, it seems worth mentioning the large reduction in the heterogeneity in specificity evident in Figure 5.	Amended so now reads "Sensitivity analysis in which only ELISA studies were included showed an improvement in specificity at the expense of sensitivity, and a reduction in the heterogeneity of specificity measurements (Figure 5).
Page 7 lines 45-6: It does not seem sensible to use the 95% CI from the meta-analysis as an estimate of the range of likely value across	Yes we agree, we have changed the text to read "We have meta-analysed the prevalence across the included studies and used this with its 95% CI

which the test would be performed in practice, as the 95% CI merely indicates the uncertainty in the summary estimate from the meta-analysis. A prediction interval (which will be wider unless the estimated between-study variance tau-squared is zero) would seem more appropriate, or simply the range of prevalences from the included studies.	as a guide to the approximate prevalence in which the tests would be performed in practice.”
Discussion:	
Page 8 lines 14-15: "The sensitivity analyses indicated that the variation seen in the Forest plots and ROC space could not be explained by test type" - this appears inconsistent with page 7 lines 21-3 and Figure 5 in which restricting to ELISA studies does explain some of the heterogeneity.	We have changed the sentence to the following "The sensitivity analyses indicated that much of the variation seen in the Forest plots and ROC space could not be explained by our measures of test type and population.”
Page 8 lines 42-5: " clinicians are likely to be interested in a combined assessment of anti-TNF levels and antibodies to anti-TNF, for which limited accuracy data is available.[21 25 43]". If this is likely to be of interest to clinicians, would it be worth giving a brief narrative summary of the results of these three studies?	We have added study outcomes from these three studies to the results on page 9 and 10.
Figure 1: The PRISMA diagram reports the number of "Full-text articles excluded with reasons", but I cannot see where the reasons were reported - if in one of the Supplements, this should be noted in the caption.	This has been added to Figure caption 1 and the list of studies excluded with reason appended as Supplement 6 .
Figures 2-3: Legends should start (or include) "Paired forest plots for..." and indicate which group of studies is Infliximab and which Adalimumab.	Amended as suggested: Figure 2 Paired forest plots for anti-TNF antibody levels for predicting loss of response or failure to regain response to Infliximab (top) and Adulimumab (bottom) Figure 3 Paired forest plots for trough anti-TNF levels for predicting loss of response or failure to regain response to Infliximab (top) and Adulimumab (bottom) The abbreviations have also been added to the legends.

Figures 4-5: The distinction between the separate panels should be explained in the legends.	Legends altered to the following: Figure 4 Hierarchical meta-analysis of trough Infliximab levels for predicting loss of response or failure to regain response. Left = all 11 studies, right = responder studies only (n = 9). The square symbol represents the summary point estimate on the HSROC curve Figure 5 Hierarchical meta-analysis of trough levels of antibodies to Infliximab for predicting loss of response or failure to regain response Top Left = all 20 studies, top right = ELISA studies only (n = 9), lower left all studies minus two influential studies (n=18),[17 25] lower right = responder studies only (n=13). The square symbol represents the summary point estimate on the HSROC curve.
Figure 6: The caption needs expanding to explain the solid and dashed red and blue lines.	Amended as follows Figure 6 Positive and negative predictive values according to prevalence of lack of response using the pooled summary ROC model estimates of sensitivity and specificity Data points = PPV and NPV at sROC pooled sensitivity and specificity and pooled prevalence. Vertical dashed lines = pooled prevalence and 95% CIs. Thick curves = PPV and NPV at upper and lower CIs for sensitivity and specificity across the pooled prevalence and its 95% CI. The dashed line ellipses encompass predictive values determined from 95% CIs of prevalence and 95% CI for PPV and NPV at the point prevalence estimate
Supplement 4 Table S4: Please indicate somewhere (e.g. in a footnote to the table) which studies were considered outliers and omitted from the second meta-analysis of trough level of antibodies to Infliximab.	Added *Outliers are Ainsworth 2008 and Steenholdt 2014

Figure S1: In the legend, "Chui" should be "Chiu". A reason should be given for omitting the study of Mazor (particularly large/influential?)	Thank you typo corrected. Added: "Mazor was omitted as it was a particularly large and influential study."
Reviewer: 2 Reviewer Name: Remo Panaccione Institution and Country: University of Calgary, Canada Competing Interests: None	
Although this is an excellent attempt to summarize the validity, sensitivity, and specificity of TDM primarily in patients who lose response to anti-TNF (namely infliximab and adalimumab) the results are not clinically relevant primarily because of the significant heterogeneity in the assays used (cut-off, sensitivity for antibody detection, range of levels) as well as the heterogeneity in the corresponding clinical definitions. Even this type of meta-analysis can't correct for this and therefore this work adds little to what is already known in the field.	Thank you for this feedback. We believe this does add considerably to the field as it is the first study to show test accuracy which is a metric of more use to clinicians than the previously published relative risk. This is because it can tell them if they use the test in clinical practice what proportion of the time it will give an incorrect answer. We agree that there is significant heterogeneity, this in itself is one of the key findings.
Reviewer: 3 Reviewer Name: Luisa Guidi MD PhD Institution and Country: Dept. Gastroenterology Fondazione Policlinico Universitario A. Gemelli, Rome, Italy Competing Interests: None declared	
The systematic review and meta-analysis is correctly designed and reported. It addresses a very significant topic in the field of inflammatory bowel disease care, that is the role of therapeutic drug monitoring of anti-TNF drugs. I have much appreciated that the Authors have accounted for all the more relevant issues in the field, the most important being the heterogeneity of assay methods and the wide range of different cut-off levels employed in the different studies. The main result of the study, that the positive and negative predictive values of these assays for loss of response or failure to regain response range between 70 and 80%, is consistent with the current opinion of the experts in the field.	Excellent, thank you.

However, this is not the only way to look at these data. The combined assay of anti-TNF and anti drug antibodies adds value, as well as the analysis of their role as a guide to manage the loss of response in a more rational way as compared to the empirical one. This could also allow a significant cost reduction. These arguments are correctly included in the discussion of this paper, as well as the acknowledgement of the need of test-treat studies to define the real clinical utility of these tests.	We have added study outcomes from these three studies to the results on page 9 and 10.
Reviewer: 4 Reviewer Name: Dr Gordon W. Moran Institution and Country: Clinical Associate Professor and Honorary Consultant Gastroenterologist, Academic Program Director for Gastroenterology, Inflammatory Bowel Disease Clinical Lead, NIHR Biomedical Research Unit in Gastrointestinal and Liver Diseases at Nottingham University Hospitals NHS Trust and The University of Nottingham, UK Competing Interests: None	
Well written paper and overall will be a valuable contribution within this field. There are some issues to target here first: A few comments that may be passed on to the authors.	Thank you.
1. The search was done in 2014. Can the authors justify this delay. A vast array of work has been published since.	We have updated the search, see our reply above.
2. Most commercial kits will not provide an antibody level if a trough level is available. I see the authors have done a sub-analysis excluding ELISAs from the analysis to investigate the effect of this. this limitation should be discussed.	By way of clarification, we undertook a sensitivity analysis which only included studies using ELISA as most studies used an ELISA test. This sensitivity analysis was undertaken to investigate whether heterogeneity could be explained by test type.
3. Why no data on ulcerative colitis has been included in this study. I appreciate the data is poor here and no trough levels has been found to be predictive.	The target patient population for this study is those with Crohns disease, though we did include studies with a small number of UC patients. The addition of ulcerative colitis studies would have added yet more heterogeneity. We agree that this would be an interesting avenue of future study. To clarify we have added to the methods “We included studies of patients with Crohn’s disease treated with Infliximab or Adalimumab. Studies with mixed Crohn’s UCI populations were

	included if the proportion of Crohn's patients was at least 70%.”
4. Are the TNF levels studied here all trough levels or are they at different stages in the dosing cycle - this needs to be discussed	Only one study used anti-TNF levels other than trough levels. We added the following to the Discussion: Test performance is dependent on ... and on the time of testing. However... . Furthermore, time of testing was not investigated as all but one study [46] reported that anti-TNFs levels considered in the studies were trough levels.
5. Methodology is commendable and nil to discuss here.	Thank you
6. The authors have described response as symptoms based - CDAI or PGA. I appreciate this is just reflective of the paucity of data published. This is a major limitation. The most predictive outcomes in IBD is mucosal healing and the dichotomous should be based on that. Suggest either doing a sub-analysis on such studies or discuss this major limitation in the literature.	We do not feel this is a major limitation but rather reflects that there is no agreed consensus as to 'best outcome', (See Alissa j Nat Rev Gastro Hepatology 2016; 13: 567-579 for a full exploration of this issue). Moreover, there is only one available study that used mucosal healing as the outcome, so we cannot do further sub-group analyses on this basis. Due to the practical issues in using this as an outcome we do not expect there to be a large number of future studies using this outcome either. However we have added to the discussion “Further there was significant heterogeneity between studies in the test used, how the outcome was assessed and unexplained variability in the findings of different studies making clinical interpretation difficult.”